# Heavy Metals as a Factor Increasing the Functional Genetic Potential of Bacterial Community for Polycyclic Aromatic Hydrocarbon Biodegradation

**DOI:** 10.3390/molecules25020319

**Published:** 2020-01-13

**Authors:** Justyna Staninska-Pięta, Jakub Czarny, Agnieszka Piotrowska-Cyplik, Wojciech Juzwa, Łukasz Wolko, Jacek Nowak, Paweł Cyplik

**Affiliations:** 1Institute of Food Technology of Plant Origin, Poznan University of Life Sciences, Wojska Polskiego 31, 60-624 Poznań, Poland; justyna.staninska@up.poznan.pl (J.S.-P.); agnieszka.piotrowska-cyplik@up.poznan.pl (A.P.-C.); jacek.nowaktz@up.poznan.pl (J.N.); 2Institute of Forensic Genetics, Al. Mickiewicza 3/4, 85-071 Bydgoszcz, Poland; pubjc@igs.org.pl; 3Department Biotechnology and Food Microbiology, Poznan University of Life Sciences, Wojska Polskiego 48, 60-627 Poznan, Poland; wojciech.juzwa@up.poznan.pl; 4Department of Biochemistry and Biotechnology, Poznan University of Life Sciences, Dojazd 11, 60-632 Poznań, Poland; lukasz.wolko@up.poznan.pl

**Keywords:** hydrocarbon biodegradation, polycyclic aromatic hydrocarbons, exopolysaccharides, microbial community

## Abstract

The bioremediation of areas contaminated with hydrocarbon compounds and heavy metals is challenging due to the synergistic toxic effects of these contaminants. On the other hand, the phenomenon of the induction of microbial secretion of exopolysaccharides (EPS) under the influence of heavy metals may contribute to affect the interaction between hydrophobic hydrocarbons and microbial cells, thus increasing the bioavailability of hydrophobic organic pollutants. The purpose of this study was to analyze the impact of heavy metals on the changes in the metapopulation structure of an environmental consortium, with particular emphasis on the number of copies of orthologous genes involved in exopolysaccharide synthesis pathways and the biodegradation of hydrocarbons. The results of the experiment confirmed that the presence of heavy metals at concentrations of 50 mg·L^−1^ and 150 mg·L^−1^ resulted in a decrease in the metabolic activity of the microbial consortium and its biodiversity. Despite this, an increase in the biological degradation rate of polycyclic aromatic hydrocarbons was noted of 17.9% and 16.9%, respectively. An assessment of the estimated number of genes crucial for EPS synthesis and biodegradation of polycyclic aromatic hydrocarbons confirmed the relationship between the activation of EPS synthesis pathways and polyaromatic hydrocarbon biodegradation pathways. It was established that microorganisms that belong to the *Burkholderiales* order are characterized by a high representation of the analyzed orthologs and high application potential in areas contaminated with heavy metals and hydrocarbons.

## 1. Introduction

Hydrocarbon compounds are currently considered as one of the most significant power sources for human activities. Development of the petrochemical industry results in the unintentional release of petroleum derivatives into natural ecosystems [1,2,3]. It is estimated that up to 8.8 million cubic meters of this energy resource is introduced annually into the environment due to accidental spills [4].

Due to the high ecotoxicity of hydrocarbon compounds and their adverse impact on human health, increasing attention has been paid to the development of remediation technologies of contaminated areas. Among them, biological methods based on the enzymatic potential of microbial consortia to metabolize the xenobiotics are very popular [5,6,7,8,9,10,11]. Bioremediation methods are characterized by low process costs and allow for the complete mineralization of xenobiotics. Additionally, they do not significantly affect the natural environment and are also effective for the removal of low concentrations of pollutants [1,9]. The difficulty in optimizing this technology due to the high impact of environmental factors is the main limiting factor in terms of efficiency [2,9,11,12,13].

The presence of heavy metals is considered as a factor that significantly affects the activity of microbial communities and their ability to metabolize hydrocarbons [14,15]. These elements enter the environment and accumulate as a result of various anthropogenic activities such as mining, energy production, electroplating, and agriculture [1,16,17]. High concentrations of heavy metals may disrupt cell membranes, affect the specificity of enzymes, inhibit the functioning of metabolic pathways in cells, and destroy the DNA structure [18]. Heavy metal contamination can negatively affect the bioremediation of hydrocarbon compounds due to the synergistic cytotoxic effect of both contaminants [18,19].

The current state of the art does not define the mechanisms regarding the influence of heavy metals on the formation and metabolic properties of bacterial populations capable of hydrocarbon degradation. Studies regarding pure strains of microorganisms have indicated that heavy metals contribute to the increase of exopolysaccharide (EPS) secretion [20,21,22]. It is believed that EPS can affect the interaction between hydrophobic hydrocarbons and microbial cells, and contribute to their increased biodegradation rate [23,24]. To date, however, there is a lack of reports that have unequivocally confirmed the relationship between the increase in EPS secretion and the frequency of the occurrence of genes involved in hydrocarbon degradation pathways in bacterial communities. Obtaining information regarding the activity of these two metabolic pathways in individual groups of an environmental metapopulation can be extremely useful for the selection of consortia used for the bioaugmentation of areas co-contaminated with heavy metals and hydrocarbon compounds.

Here, we report the effect of heavy metals on the environmental microbial community structure, activity, and predicted frequency of the occurrence of genes involved in hydrocarbon degradation and EPS secretion.

## 2. Results

### 2.1. Hydrocarbon Biodegradation

The analysis of the kinetics of hydrocarbon dissipation confirmed the significant impact of heavy metals on the biological degradation processes (Figure 1). Strong inhibition was noted in the case of the biodegradation of the aromatic hydrocarbon fraction of diesel oil (decreased by 54.3% for M50 and 65.3% for M150 when compared to the control after 168 h of the experiment). Similar trends were observed for the aliphatic hydrocarbon fraction (decreased by 50.7% for M50 and 63.2% for M150 after 168 h of the experiment). In the group of straight-chain aliphatic hydrocarbons, a long-lasting inhibition effect was observed only in the case of the M150 variant, which contained a higher concentration of heavy metals. The positive effect of heavy metals on the biodegradation efficiency of the polycyclic aromatic hydrocarbon (PAH) fraction, a group of compounds that was the most resistant to the biodegradation by the analyzed microbiological consortium, was also noted. In this case, the M50 and M150 variants exhibited an increase in biodegradation efficiency by 17.9% and 16.9%, respectively.

### 2.2. Metabolic Activity of Microorganisms

The presence of heavy metals negatively affected the microbial community (Figure 2). The percentage ratio of the active population Q2 in variants M50 and M150 after 24 h of the experiment was equal to 35.92% (± 0.52%) and 34.57% (± 0.47%), respectively, which was a significant decrease compared to the control sample (Q2 = 98.60% ± 0.14%). The effect of toxicity persisted throughout the whole biodegradation process. After 168 h, the following Q2 values of 35.80% ± 1.18% and 39.87% ± 0.24% were noted for the M50 and M150 variants, respectively, in relation to the control sample Q2 = 93.60% ± 0.15%.

### 2.3. Taxonomic Structure and Biodiversity

The influence of heavy metals on the taxonomic structure of microbial consortia is presented in Figure 3 and Figure 4. A decrease in the ratio of *Alphaproteobacteria, Sphingobacteria*, and *Flavobacteria*, and an increase in the ratio of *Gammaproteobacteria, Betaproteobacteria*, and *Bacilli* class relative to the control test were noted. Classification to the level of order indicated a high decrease of the ratio of *Sphingobacteriales* (by 21.6%) and *Rhizobiales* (by 19.1%) as well as an increase in the ratio of *Xanthomonadales* (by 23.2%), *Burkholderiales* (by 12.7%), and *Sphingomonadales* (by 5.9%). The *Xanthomonadales* and *Burkholderiales* orders were the dominant taxa (30.6% and 30.4% of the population, respectively).

The values of the analyzed alpha-biodiversity coefficients in the metapopulation of microorganisms after 168 h of the biodegradation experiment are presented in Table 1. The presence of heavy metals negatively affected the metapopulation richness (number of operational taxonomic unit (OTU) and Chao1) and phylogenetic diversity.

### 2.4. Phylogenetic Investigation of Communities by Reconstruction of Unobserved States (PICRUSt) Analysis

#### 2.4.1. Gene Orthologs Participating in Hydrocarbon Biodegradation

The results of the analysis are presented in Figure 5. A significant effect of the presence of heavy metals on the predicted number of gene copies and their representation in individual taxa was established, while a positive effect was found in the case of the *nahAb*, *nahAc*, and *pht5* orthologues. In the group of analyzed KEGG Orthologies (KO), microorganisms that belong to the *Burkholderiales* order were characterized by the highest predicted number of copies in metapopulations. 

#### 2.4.2. Gene Orthologs Participating in Exopolysaccharides Synthesis

The presence of heavy metals contributed to the increased number of copies of most genes involved in EPS synthesis (except for *tagA* and *wcaJ*). Similarly, for the majority of analyzed KOs, an increase in the predicted copy number of genes was observed for the *Burkholderiales* and *Xanthomonadales* orders (Figure 6). These groups were also the dominant taxa in the representation of all KOs, except for *tagA*.

## 3. Discussion

The studies indicate that the consortium with high potential for the biodegradation of hydrocarbons was highly sensitive to heavy metals. The observed decrease in metabolic activity relative to the control sample, both after 24 and 168 h of the biological decomposition process, showed their negative effect on the microbial cells. There was also a decrease in the metapopulation richness expressed by the OTU number, Chao1 index, and phylogenetic diversity coefficient. Similar results were obtained by Xie et al. (2016) during an investigation of a consortium of soil microorganisms [25]. The authors reported that biological activity and biodiversity decreased as the concentration of heavy metals increased. Analyses carried out by Šmejkalová et al. (2003) indicated a similar relationship [26]. Decreased biodiversity has also been reported in the studies by Chodak et al. (2013) regarding the impact of lead, zinc, and cadmium on soil microflora [27]. The decrease in metabolic activity and biodiversity of the microbiological consortium in the presence of heavy metals can be explained by the mechanisms of their toxic effects on microbial cells. Such mechanisms include the phenomena of free radical formation in the intracellular environment, inactivation of enzymes, influence on their conformational structure, complexation of thiol-derived components of the cell membrane and its degeneration, chelation by key cell metabolites, and competition between metal ions and other key ions and compounds in cells, which are described relatively well in the literature [28,29]. Under the influence of heavy metals, there is a reduction in the number and diversity of microorganisms with low tolerance, which do not possess the metabolic adaptations to proliferate at high concentrations of such contaminants [29]. A similar observation was reported by Yao et al. (2003), emphasizing that the aforementioned phenomenon of selection may result in the development of a population of microorganisms with characteristics that enable the reduction of the negative effects of heavy metals [30].

Analysis of changes in the taxonomic structure of the microbial consortium indicates the low tolerance of the *Sphingobacteriales* order, while the resistance of the *Xanthomonadales* and *Burkholderiales* orders was relatively high. It can therefore be assumed that the microorganisms that belong to the *Xanthomonadales* and *Burkholderiales* orders have developed mechanisms to overcome the negative effects of heavy metals. The presence of *Xanthomonadales* and *Burkholderiales* was also reported in the research of Kou et al. (2016), which focused on the analysis of sediment microflora of Lake Poyang (China) characterized by a high concentration of heavy metals [31]. Among the most important mechanisms of tolerance to heavy metals is the phenomenon of intracellular sequestration (accumulation of metal ions in the cytoplasm by binding to proteins), extracellular sequestration, enzymatic detoxification, modification of cell membrane permeability, removal of xenobiotics by active transport through membranes, and DNA repair mechanisms. Genes that determine the response to stress associated with the presence of heavy metals can be encoded both chromosomally and by plasmids [32]. In the framework of the experiment, which is the topic of this study, M150 variants exhibited a slight increase in metabolic activity after 168 h of biodegradation. This can be explained by both the increase of the percentage ratio of taxa characterized by high resistance to heavy metals and the possible horizontal transfer of genes associated with this resistance. Previous studies by Tewari et al. (2013) have highlighted the high efficiency of the phenomenon of plasmid transfer in the context of resistance to these xenobiotics [33].

The analysis of the biodegradation kinetics of total hydrocarbons indicated numerous limitations of the biological remediation process of areas rich in heavy metals, in the case of both aliphatic and aromatic hydrocarbon fractions. One of the exceptions in the negative impact of heavy metals on the efficiency of hydrocarbon biodegradation is the fraction of n-alkanes, which, in the presence of lower concentrations of heavy metals (50 mg L^−1^), did not show a significant effect on their biodegradation after 168 h of the process, despite the initial inhibition. It can be assumed that this is associated with the process of the formation and subsequent stabilization of the bacterial metapopulation. In the first days of the experiment, the total number of microorganisms, and thus the total biodegradation potential, decreased as a result of the selection factor. After 168 h, the environmental niche was filled by growing populations capable of living in the presence of heavy metals. It is believed that n-alkanes are a fraction readily degraded by the majority of microorganisms capable of degrading hydrocarbons [34], therefore, the mentioned selection did not ultimately reduce the potential for enzymatic degradation of this group of compounds. The second exception is the group of polyaromatic compounds, in this case, there was an increase in the efficiency of the biological decomposition for both tested concentrations (50 mg·L^−1^ and 150 mg·L^−1^). The effect of the increased rate of polycyclic aromatic hydrocarbon degradation in the presence of metals has been previously described in the literature. Researchers have observed the positive effect of manganese on the biological degradation of phenanthrene, fluorene, and fluoranthene. The mechanism of interaction has not been clearly defined [17] but is most likely associated with the phenomenon of the increased activity of enzymes crucial for the biological degradation of polyaromatic compounds, dioxygenases. The results obtained in the framework of this study confirm this hypothesis. In the analyzed group of genes, an increase in the expected number of copies of genes encoding the naphthalene dioxygenase enzyme (*nahAb* and *nahAc*) was noted, while in the majority of orthologs involved in the subsequent stages of metabolic degradation, such a relationship was not found. Analysis of the number of gene copies encoding enzymes that initiate the PAH biodegradation process such as dioxygenases can be extremely helpful for assessing the potential of the microbial community to degrade these compounds. A similar phenomenon of the increased rate of PAH biodegradation by an environmental consortium in the presence of heavy metals was observed in an earlier work by Czarny et al. (2020) [35]. Interestingly, the presence of heavy metals also contributed to the increase of the relative representation of most KOs by microorganisms that belonged to the *Burkholderiales* order. Considering the fact that this group is one of the dominant taxa in the metapopulation, it can be established that it is characterized by a high application potential in the bioremediation of areas with high concentrations of polyaromatic hydrocarbons.

The results of the experiment suggest that the phenomenon of increased dissipation efficiency of polyaromatic hydrocarbons can be associated with increased secretion of cellular exopolymers. In addition to the high representation of genes encoding naphthalene dioxygenase as described above, microorganisms that belong to the *Burkolderiales* order were also characterized by a high number of the estimated copy of most genes involved in the synthesis of cellular exopolymers. This phenomenon could result from the progressive dominance of species with well-developed mechanisms that allowed for the elimination of the negative effects of heavy metals through their immobilization. According to Gupta and Diwan (2017), the synthesis of EPS is one of the reactions of microorganism cells to the presence of heavy metals [36]. This was also confirmed by numerous studies by other researchers [37,38]. Wang et al. (2010) explains the phenomenon of sorption by the interaction between positively charged metal ions and negatively charged EPS [39]. The increased efficiency of PAH degradation can therefore be associated with the impact of EPS, which increased the interaction of PAH with the *Burkolderiales* cells. The ability of EPS to solubilize hydrophobic contaminants was confirmed by other studies [24,40]. It can be assumed that the addition of factors that stimulate the development of the metapopulation structure toward the dominance of taxa capable of EPS synthesis such as heavy metals is an effective method of selection consortia used for bioremediation of PAH contaminated areas.

## 4. Materials and Methods 

### 4.1. Microbial Community Enrichment

The environmental bacterial consortium was isolated from soils permanently contaminated with petroleum compounds from the areas of the former saturation plant of railway sleepers in Solec Kujawski (Poland): 53°04′40.7′′N, 18°14′23.6′′E. The soil samples were deposited in May 2017 (air temperature above 25 °C). The pH of the soil was neutral. Functional screening was carried out in order to obtain a population characterized by a high potential for the degradation of hydrocarbon compounds. Microorganisms were cultivated using 50 mL of mineral salt medium (MSM) described by Szczepaniak et al. (2015) [41], which included: Na_2_HPO_4_ 6.21 g L^−1^, KH_2_PO_4_ 2.8 g L^−1^, NaCl 0.5 g L^−1^, NH_4_Cl 1.0 g L^−1^, and 200 µL trace element solution (TES): MgSO_4_ × 7H_2_O 0.01 g L^−1^, FeSO_4_ × 7H_2_O 0.001 g L^−1^, MnSO_4_ × 4H_2_O 0.0005 g L^−1^, ZnCl_2_ 0.00064 g L^−1^, CaCl_2_ × 6H_2_O 0.0001 g L^−1^, BaCl_2_ 0.00006 g L^−1^, CoSO_4_ × 7H_2_O 0.000036 g/L, CuSO_4_ × 5H_2_O 0.000036 g/L, H_3_BO_3_ 0.00065 g/L, H_2_MoO_4_ 0.005 g/L, EDTA 0.001 g L^−1^, and HCl 37% 0.0146 mL L^−1^. Additionally, 2 g of diesel oil (PKN Orlen, Płock, Poland) was added to the mineral medium, which was the sole carbon source. The cultures were cultivated for five days under aerobic conditions at 25 °C and neutral pH. The obtained biomass was centrifuged (10 min, 4000 rpm) and suspended in physiological saline. The number of microbial cells was normalized based on the measurement of optical density to OD (600) = 0.7 (Helios Delta Vis, ThermoFisher Scientific, Waltham, MA, USA). The resulting suspension was used as the inoculum for the biodegradation experiment.

### 4.2. Biodegradation Experiment

The biodegradation experiment was carried out in flasks with baffles and a stopper with a semi-permeable DURAN^®^ membrane. The culture medium consisted of 50 mL of MSM mineral medium, 200 µL of TES solution, 2 g of diesel oil (PKN Orlen), and heavy metal salts: PbCl_2_ (Sigma-Aldrich, St. Louis, MI, USA), CdCl_2_ (Sigma-Aldrich), NiCl_2_ (Sigma-Aldrich), ZnCl_2_ (Sigma-Aldrich), CuCl_2_ (Sigma-Aldrich), and CrCl_3_ (Sigma-Aldrich). The systems contained a final concentration of 8.33(3) mg·L^−1^ (variant designated as M50) and 25 mg·L^−1^ (variant designated as M150 of each of the heavy metals. The cooperative effect of heavy metals was tested. The total amount of heavy metals in samples M50 and M150 was 50 mg·L^−1^ and 150 mg·L^−1^, respectively. The systems were inoculated with 250 μL of the microbial inoculum. Biodegradation was carried out for seven days under aerobic conditions at 25 °C.

### 4.3. Analysis of Hydrocarbon Dissipation

The dissipation of specific hydrocarbon groups was assessed after 24, 72, and 168 h of biodegradation in relation to the initial concentration of compounds. Approximately 2.5 mL of acetone (POCH, Gliwice, Poland) was added to each experimental flask and shaken for 10 min (150 rpm). Then, 200 μL of an internal standard solution were dosed: nonane (Sigma-Aldrich) 6 mg/mL, octacosane (Restek, Lisses, France) 6 mg∙mL^−1^ and acenaphthene-d10 (Restek) 1.5 mg mL^−1^ in a acetone/hexane mixture (2:3 *v/v*). After mixing, 25 mL of hexane (POCH) was added to the flasks. The systems were shaken for 60 min (150 rpm) and left for 45 min to separate the phases. Then, 2 mL of acetone (POCH) were added to eliminate any emulsion. The systems were stabilized for 20 min, then the extracts were diluted. A small portion of anhydrous MgSO_4_ (Chempur, Piekary Śląskie, Poland) was added to the chromatographic vials to bind residual water, followed by 960 mL of hexane (POCH), and 40 mL of the extract. The obtained samples were stored at 4 °C. Immediately before the chromatographic analysis, the extracts were diluted 25-fold with hexane (POCH). The gas chromatography–mass spectrometry analysis was performed using a Shimadzu 17A gas chromatograph coupled with a QP5000 mass spectrometer (Canby, OR, USA). The device was equipped with an Rxi-5MS column (30 m long, 250 µm in diameter, 0.25 µm stationary phase thickness) connected to a 2.5 m long pre-column (Restek). The analysis conditions were as follows: carrier gas, helium; flow, 1.1 mL∙min^−1^; injector temperature, 250 °C; injected sample volume, 1 µL; temperature program, initial temperature at 40 °C maintained for 1 min, followed by an increase by 15 °C∙min^−1^ up to 300 °C, which was maintained for 7 min. The detector performed a scan every 0.15 s in the 33/250 *m/z*. Each sample was analyzed in triplicate.

### 4.4. Analysis of Metabolic Activity

The analysis of metabolic activity was based on the procedure described by Cyplik et al. (2013) [42]. Approximately 1 mL of the culture medium was taken for analysis after 24 and 168 h of incubation. The samples were centrifuged (5 min, 3500 rpm) to separate the microbial cells. After decanting the supernatant, the pellet was washed twice using 1 mL of phosphate buffered saline (Sigma-Aldrich) and centrifuged (5 min, 3500 rpm). Then, the cells were suspended in 250 μL of phosphate buffered saline (Sigma-Aldrich) and 1 μL of Redox Sensor Green dye from the BacLight ™ RedoxSensor ™ Green Vitality Kit (Thermo Fisher Scientific, Waltham, MA, USA) was added to each sample. The systems were incubated for 10 min at room temperature. Fluorescence in the stained samples was measured using a BD FACS Aria ™ III flow cytometer (Becton Dickinson, Franklin Lakes, NJ, USA). The percentage ratio of active population (Q2) was determined by gating point plots of median values of green fluorescence signals (FITC-A) versus medians of light scattering values (SSC-A) based on a comparison to a previously prepared negative sample that contained metabolically thermally inactivated dead cells.

### 4.5. Analysis of Changes in the Taxonomic Structure

#### 4.5.1. Isolation of DNA

Isolation of genomic DNA was carried out using the Genomic Mini AX Bacteria Spin Kit (A&A Biotechnology, Gdynia, Poland), according to the protocol provided by the manufacturer. The isolation efficiency was checked each time by fluorimetry using a Qbit 3.0 device and the Qubit ™ dsDNA HS Assay Kit (ThermoFisher Scientific). Three extractions of DNA were performed for each sample, which were finally combined after positive quantitative assessment.

#### 4.5.2. Polymerase Chain Reaction (PCR) Amplification and Sequencing

The Vs 16S rRNA region was amplified based on the 515F-806R primers designed by Caporaso et al. (2012) [43]:

F:AATGATACGGCGACCACCGAGATCTACACTATGGTAATTGTGTGCCAGCMGCCGCGGTAA

R:CAAGCAGAAGACGGCATACGAGATXXXXXXAGTCAGTCAGCCGGACTACHVGGGTWTCTAAT

The PCR mixture consisted of 0.25 μg of genomic DNA, 0.5 μM of each of the primers, 2.5 μL of nuclease-free water (ThermoFisher Scientific), and 12.5 μL of a PCR Master Mix Kit with the Taq polymerase (ThermoFisher Scientific). The reaction was carried out using an ABI 2724 device (Applied Biosystem, Waltham, MA, USA). The following temperature program was applied: initial denaturation for 3 min, 95 °C, 35 cycles: denaturation 1 min, 95 °C; attachment of primers for 30 s, 52 °C; elongation for 1 min 72 °C, final elongation for 10 min 72 °C. The PCR products were purified using a Clean-Up columns set (A&A Biotechnology), according to the procedure provided by the manufacturer. Sequencing of the obtained amplicons was carried out using the MiSeq (Illumina, CA, USA) platform. The amplicons were constructed based on the recommended NEBNext^®^ DNA Library Prep Master Mix Set for Illumina (New England Biolabs, Ipswich, MA, USA), according to the manufacturer’s protocol. The libraries were normalized to equimolar concentrations and quantified by fluorimetry using a Qubit 3.0 (Invitrogen, Waltham, MA, USA) device and the dsDNA HS assay kit (Life Technologies, Camarillo, CA, USA). Then, they were denatured in the presence of 0.2 N NaOH prepared earlier and diluted using a HT1 (Illumina) buffer to a final concentration of 8 pM. The libraries were sequenced using an Illumina MiSeq device (Illumina) and the MiSeq Reagent Kits v2 kit (Illumina). The primers proposed in the study of Caporaso et al. (2012) were used. The sequencing datasets generated and analyzed during the current study are available in the SRA repository, under the identifier BioProject PRJNA591765.

#### 4.5.3. Bioinformatic Analysis

The output data for the sequencing process in FASTQ format were imported into the CLC Genomics Workbench 8.5 software with the CLC Microbial Genomics Module 1.2 (Qiagen, Hilden, Germany). The reads were demultiplexed and paired ends were connected (mismatch factor = 2, minimum value factor = 8, maximum gap size = 3, maximum end mismatch = 5). Then, the primer sequences were trimmed (qualitative limit = 0.05, ambiguity limit = ‘N’) and the identification and removal of chimeric readings were performed. The output data were clustered independently using two reference databases: SILVA v119 [44] and GreenGenes 13.5 [45], at the level of 97% similarity of operational taxonomic units (OTU). Biodiversity coefficients: OTU number, Chao1 coefficient, and phylogenetic diversity were determined based on the tabular listing of the frequency of defined OTUs (clustered against SILVA v119).

### 4.6. PICRUSt Analysis

In order to determine the expected number of copies of the selected genes relevant for the biodegradation processes of polycyclic aromatic hydrocarbons and exopolysaccharide secretion (EPS) in bacterial metapopulations, the bioinformatic tool PICRUSt (Phylogenetic Investigation of Communities by Reconstruction of Unobserved States) developed by Langille et al. (2013) was used [46]. This algorithm enabled the prediction of the metapopulation functional content and contribution of OTUs to the abundance of each function based on the genomic sequencing data of the 16S rRNA fragment. Based on the Kyoto Encyclopedia of Genes and Genomes (KEGG) database, orthologs (KO) compatible with the PICRUSt algorithm and also key in terms of the analyzed metabolic pathways (Table 2) were selected [47]. Qualitative assessment of the analysis was based on the determination of the weighted nearest sequenced taxon index, which expressed the level of representation of identified OTUs in the Integrated Microbial Genomes & Microbiomes database [46].

## 5. Conclusions

The presence of heavy metals in the environment significantly affects the activity and metapopulation structure of bacterial consortia. Despite the reduced metabolic activity, microorganisms are able to effectively degrade one of the most problematic groups of hydrocarbon compounds: PAH. Exposure to heavy metals contributes to the increased abundance of taxa with genetic orthologs, which enable the synthesis of exopolysaccharides. EPS contribute to the increased bioavailability of hydrophobic pollutants due to their ability to solubilize hydrophobic compounds and change the hydrophobicity of cells. The high concentration of bioavailable PAH favors the growth of taxa with orthologs of genes that initiate the biodegradation process. Microorganisms that belong to the *Burkolderiales* order, due to the coexistence of orthologs responsible for both the synthesis of EPS and the biodegradation of PAH, possess great potential for the bioremediation of PAH-contaminated areas.

## Figures and Tables

**Figure 1 molecules-25-00319-f001:**
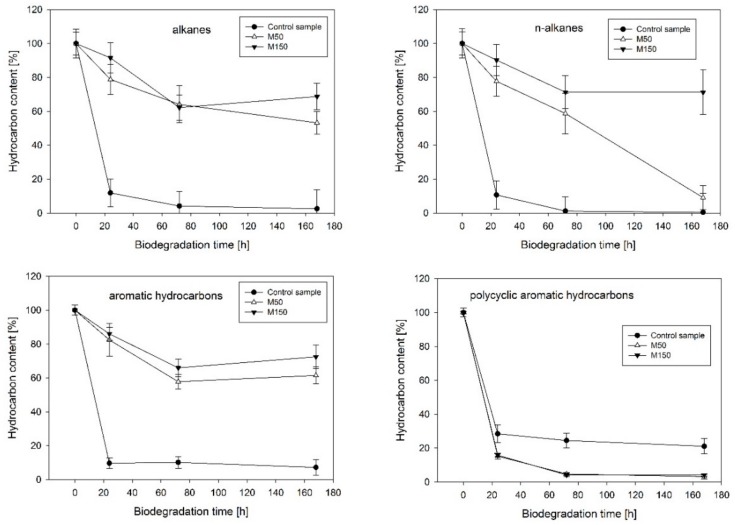
Kinetics of biodegradation of the selected hydrocarbon fractions under the presence of heavy metals (M50 and M150 correspond to 50 mg·L^−1^ and 150 mg·L^−1^, respectively).

**Figure 2 molecules-25-00319-f002:**
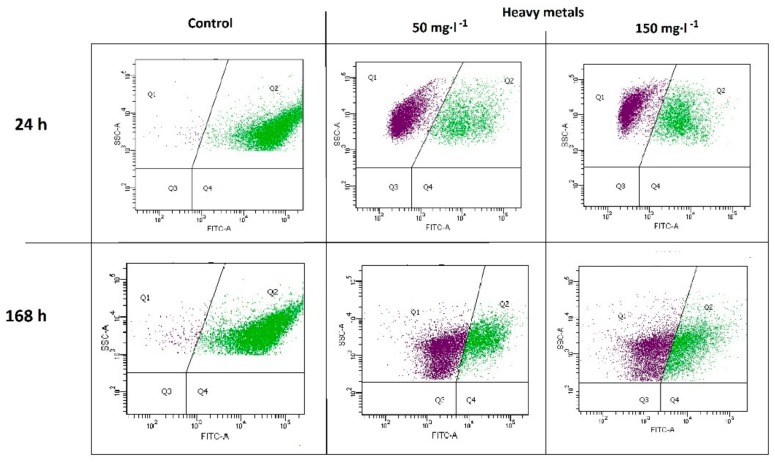
Flow cytometry data of microbial activity after 24 h and 168 h of biodegradation.

**Figure 3 molecules-25-00319-f003:**
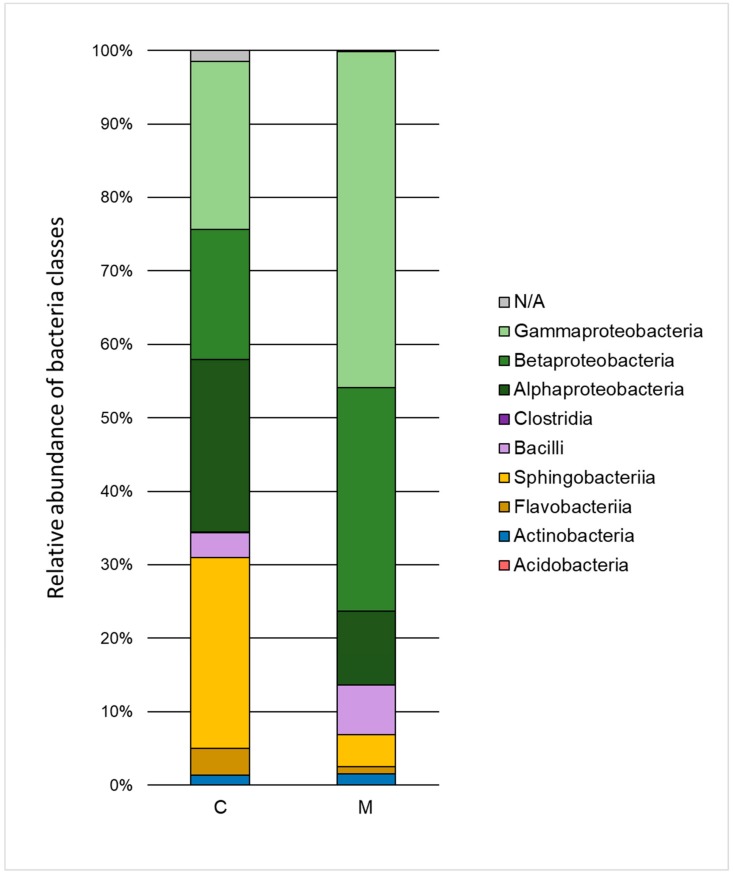
Relative abundance of bacterial classes in the control sample (C) and in a sample containing 150 mg·L^−1^ of heavy metals (M) after 168 h of the biodegradation process.

**Figure 4 molecules-25-00319-f004:**
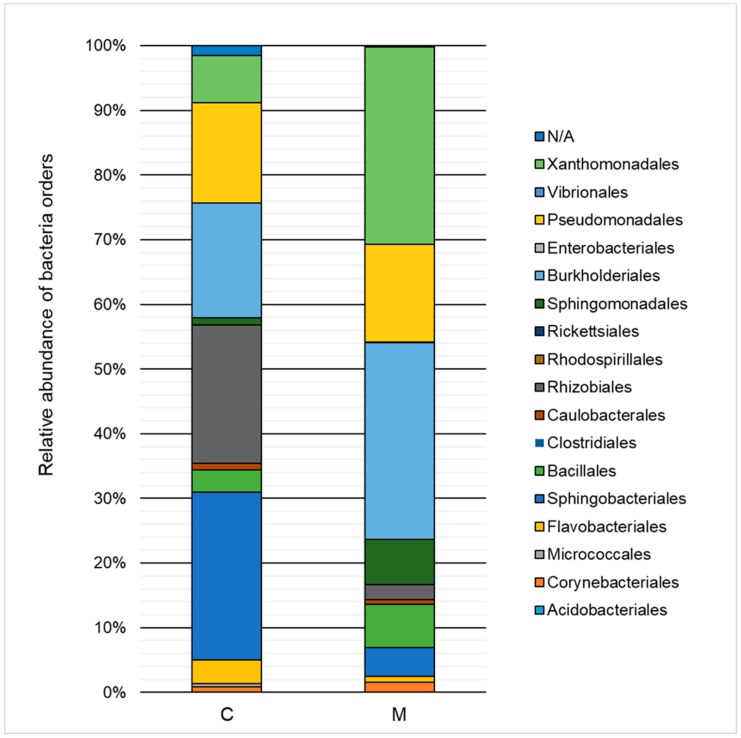
Relative abundance of bacterial orders in the control sample (C) and in a sample containing 150 mg·L^−1^ of heavy metals (M) after 168 h of the biodegradation process.

**Figure 5 molecules-25-00319-f005:**
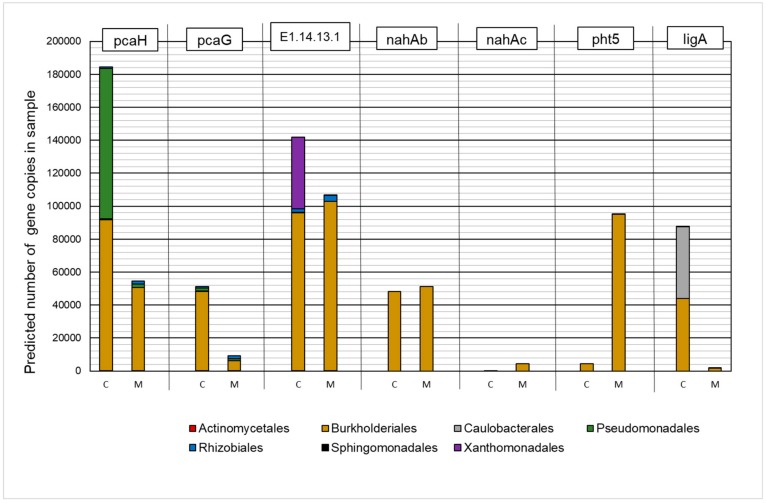
The contribution of the predicted number of gene copies involved in polycyclic aromatic hydrocarbon biodegradation to bacteria orders in the control sample (C) and in a sample containing 150 mg·L^−1^ of heavy metals (M).

**Figure 6 molecules-25-00319-f006:**
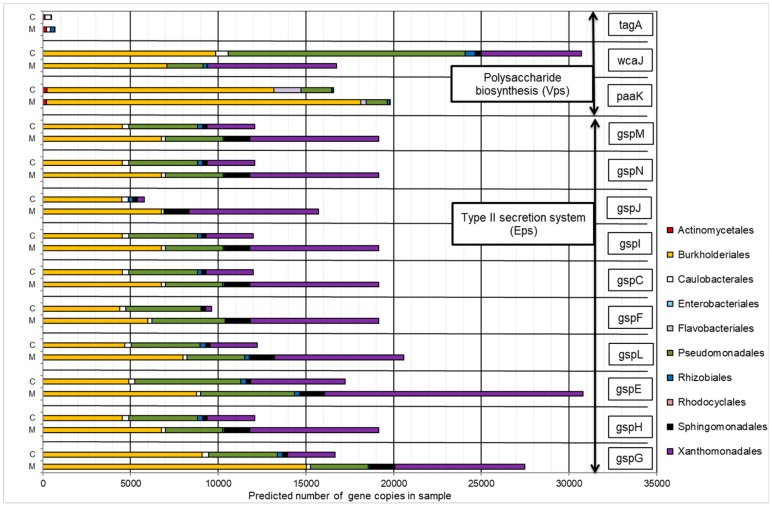
The contribution of the predicted number of gene copies involved in EPS synthesis to bacteria orders in the control sample (C) and in a sample containing 150 mg·L^−1^ of heavy metals (M).

**Table 1 molecules-25-00319-t001:** Values of the metapopulation biodiversity coefficients for microorganisms after 168 h of biodegradation.

	Number of OTU	Chao1	Phylogenetic Diversity
**Control sample**	95	107	4.41
**M150**	85	96	4.16

**Table 2 molecules-25-00319-t002:** Characterization of the analyzed Kyoto Encyclopedia of Genes and Genomes orthologs.

KO Number	Gene Name	Definition	EC Number
**PAH biodegradation**			
K00449	*pcaH*	protocatechuate 3,4-dioxygenase, beta subunit	EC: 1.13.11.3
K00448	*pcaG*	protocatechuate 3,4-dioxygenase, alpha subunit	EC: 1.13.11.3
K00480	*E1.14.13.1*	salicylate hydroxylase	EC: 1.14.13.1
K14578	*nahAb*	naphthalene 1,2-dioxygenase ferredoxin component	EC: 1.14.12.12
K14579	*nahAc*	naphthalene 1,2-dioxygenase subunit alpha	EC: 1.14.12.12
K04102	*pht5*	4,5-dihydroxyphthalate decarboxylase	EC: 4.1.1.55
K04100	*ligA*	protocatechuate 4,5-dioxygenase, alpha chain	EC: 1.13.11.8
**EPS synthesis**			
K02456	*gspG*	general secretion pathway protein G	EC: 7.4.2.8
K02457	*gspH*	general secretion pathway protein H	EC: 7.4.2.8
K02454	*gspE*	general secretion pathway protein E	EC: 7.4.2.8
K02455	*gspF*	general secretion pathway protein F	EC: 7.4.2.8
K02452	*gspC*	general secretion pathway protein C	EC: 7.4.2.8
K02453	*gspD*	general secretion pathway protein D	EC: 7.4.2.8
K02458	*gspI*	general secretion pathway protein I	EC: 7.4.2.8
K02459	*gspJ*	general secretion pathway protein J	EC: 7.4.2.8
K02463	*gspN*	general secretion pathway protein N	EC: 7.4.2.8
K02462	*gspM*	general secretion pathway protein M	EC: 7.4.2.8
K02461	*gspL*	general secretion pathway protein L	EC: 7.4.2.8
K02460	*gspK*	general secretion pathway protein K	EC: 7.4.2.8
K01912	*paaK*	phenylacetate-CoA ligase	EC: 6.2.1.30
K03646	*wcaJ*	putative colanic acid biosynthesis UDP-glucose lipid carrier transferase	EC: 2.7.8.31
K05946	*tagA*	*N*-acetylglucosaminyldiphosphoundecaprenol N-acetyl-beta-D-mannosaminyltransferase	EC: 2.4.1.187

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
