# Peer review of "Heavy Metals as a Factor Increasing the Functional Genetic Potential of Bacterial Community for Polycyclic Aromatic Hydrocarbon Biodegradation"

_molecules, 2020, doi:10.3390/molecules25020319_

Round 1

Reviewer 1 Report

The manuscript by Staninska-Pieta et al., studied the impact of heavy metals on the genetic potential of an environmental consortium for polycyclic aromatic hydrocarbon biodegradation. The authors concluded that presence of heavy metals resulted in a decrease of metabolic activity of the microbial consortium and its biodiversity but also in an increase of the biological degradation rate of polycyclic aromatic hydrocarbons. The author finally concluded that there was a relationship between the activation of EPS synthesis pathways and polyaromatic hydrocarbon biodegradation pathways. These results are of potential interest for a broad audience, specifically those involved on metal and hydrocarbon contamination and bioremediation. Overall this manuscript is well written but could benefits from some edits.

Comments:

Line 24: EPS should be defined here and not in line 28.

Lines 42-45: the term “energy source” when referring to hydrocarbon compounds and microorganisms is confusing. I guess the authors are referring to “energy source” from an anthropogenic point of view (fuels). However, the term energy source could also refer to its role supporting microbial communities. Although hydrocarbon compounds can be used as energy and carbon source by some microorganisms, by no means is one of the “most significant energy source” for microbes. Please be more specific

Lines 54-55: Please add references.

Lines 57-59: Please add references.

Line 65: EPS should be defined here, after exopolysaccharide.

Line 69: delete “the” (in bacterial communities).

Line 72-73: At the end of the introduction the authors should write a short paragraph with information regarding what are they going to show in the manuscript (“Here we reported the…..”).

Lines 75-85: The authors described the effect of heavy metals on hydrocarbon degradation, but it is not clear which heavy metal was used in the experiment. In the method section the authors said that different heavy metals where used but is its not totally clear if these results corresponded to the effect of all these heavy metals together or if the authors test the effect of each heavy metal separately. Please add more information in the method section as well.

Figure 1: Please increase the font of the axes.

Figure 1 legend: Please add more information to the legend so the readers do not have to go to the method section. (i.e. experiment were performed in triplicate, which heavy metal? etc). There are two dots after “1”, please fix this.

Lines 89-95: These results are not showed. My understanding is that these determinations were based on flow cytometry. The authors should include some images showing these differences, and if possible, some fluorescent micrograph as well.

Line 94: It should say M50 and not M59.

Lines 96-103: Why these analyses were not done with M50 as well? Why the authors only referred to the changes of some bacterial classes and not others? For example, the authors said that “decrease of the ratio of Sphingobacteria and Flavobacteria and an increase of the ratio of Betaproteobacteria class relative to the control test were noted” However, there was also a decrease of Alphaproteobacteria and an increase of Gammaproteobacteria and Bacilli, but the authors did not discuss this. The same problem was also observed for figure 3.

Lines 110-113: The same analyses should be done with M50.

Line 117: The authors should be more precise. These KO are involved in polycyclic aromatic hydrocarbons degradation. Do not use KO here, use ortholog instead.

Lines 117-120: I do not think that these results are enough to validate that heavy metals increase the functional genetic potential for polycyclic aromatic hydrocarbons biodegradation since from the 7 orthologs involved in these degradations, only 3 have a positive effect with metals.

Figure 4 and 5 legends: Please add more information to the legend so the readers do not have to go to the method section. Define C and M. Since figure 4 is related to polycyclic aromatic hydrocarbons the authors should said this instead of just “hydrocarbon”.

Line 126: Do not use KO here, use ortholog instead.

Lines 136-145: As was previously mentioned, the analyses of population richness should be also done with M50.

Line 152: Add space before [26, 27].

Line 191: PAH should be defined the first time mentioned.

Lines 196-197: Although the copy numbers of some dioxygenases genes increased in the presence of metals, this was not true for all dioxygenases. Moreover, this increment does not necessarily mean an increase in dioxygenase activity so I do not think that the authors can conclude that this study confirm this hypothesis.

Line 217: use EPS instead of exopolysaccharide.

Figure 1, 2, 3,4 and 5 legends: There are two dots after the number. Please fix.

Line 228: The authors should change “isolation” for “enrichment” since the authors do not have “pure isolates cultures”

Lines 229-230: Please add more information of the environment from where the sample was collected (pH and temperature at least).

Line 231-240: Include the pH of the culture media.

Line 235: add space after 200.

Lines 244-253: My understanding is that the cultures have all the metals together. This should be clarified.

Line 250: add space after M150.

Lines 258-259: sometimes mg/ml is used and other times mg ml-1. Please be consistent through all the manuscript.

Line 291: should say Qbit 3.0 instead of Qbit.

Line 295: should say 16S rRNA. Please fix through all the manuscript.

Lines 301-304: Instead of giving volumes, provide the final concentrations used. Volumes are useless if the concentration of the solutions are not provided.

Table 2: Please double check all genes names because some of them are misspelled (hydroxylase, decarboxylase, etc).

Reviewer 2 Report

The authors have described the effects of heavy metals on the activity of PAH degradation in oil-contaminated soil. Overall, the design of the experiment was incorrect and the method of analysis of the results was inadequate, making it unsuitable for publication in this journal. When resubmitting, the following should be improved.

PICRUSt is not a magic tool to infer the functions based on 16S rRNA. It provides very rough picture of the functions. No matter how similar the 16S genes are, there are many genera that differ significantly in function, and they should be verified through metagnome (shot-gun) analysis.  Replicates: Soil samples with complex microbial communities require at least three replicates and 16S rRNA analysis also require at least three replicates. Metabolic vs Phylogenetic: If you want to identify the assocation between metabolic activity with microbial populations, you must perform RNA-based assays or isotopic experiments. Flow cytometry provides information on the overall active population but does not provide information on which microorganisms play a major role in PAH degradation.  The role of EPS:  Without the quantitative results of EPS, it is difficult to determine the role of EPS in PAH degradation. PICRUSt results are not quantitatively available and their association with metabolic activity is more difficult to explain. PAH degradation: Although the initial results in carbohydrate decomposition experiments were mostly due to adsorption by soil particles, the adsorption effects were not fully analyzed or considered. It is difficult to agree with the author's conclusion without demonstrating that heavy metals affect the adsorption efficiency of soil particles rather than PAH decomposition efficiency.

Round 2

Reviewer 1 Report

The authors significantly improved their manuscript and responded to all my comments. Now it is easier to follow and it will be a good source for hydrocarbon biodegradation studies

Author Response

We are very grateful for Your consideration of our manuscript. 

Reviewer 2 Report

Unfortunately, the author did not reflect any comments from reviewer at all. If it is difficult to respond to the reviewer's comments based on current results or current approaches, the author should discuss why the comments was difficult to apply in the research, but did not take any action. I consider this manuscript not suitable for publication in this journal.

Author Response

We are very grateful for Your consideration of our manuscript. Please see the attached file.
